# Complementary Feeding in Preterm Infants: A Systematic Review

**DOI:** 10.3390/nu12061843

**Published:** 2020-06-20

**Authors:** Nadia Liotto, Francesco Cresi, Isadora Beghetti, Paola Roggero, Camilla Menis, Luigi Corvaglia, Fabio Mosca, Arianna Aceti

**Affiliations:** 1Fondazione IRCCS Ca’ Granda Ospedale Maggiore Policlinico, Neonatal Intensive Care Unit, 20122 Milan, Italy; nadia.liotto@policlinico.mi.it (N.L.); paola.roggero@unimi.it (P.R.); camilla.menis@gmail.com (C.M.); fabio.mosca@unimi.it (F.M.); 2Department of Clinical Sciences and Community Health, University of Milan, 20122 Milan, Italy; 3Neonatology and Neonatal Intensive Care Unit, University of Turin, 10126 Turin, Italy; francesco.cresi@unito.it; 4Neonatal Intensive Care Unit, AOU Bologna, Department of Medical and Surgical Sciences, University of Bologna, 40138 Bologna, Italy; i.beghetti@gmail.com (I.B.); arianna.aceti2@unibo.it (A.A.)

**Keywords:** complementary feeding, weaning, preterm infants, oral dysfunction

## Abstract

Background: This systematic review summarizes available literature regarding complementary feeding (CF) in preterm infants, with or without comorbidities that may interfere with oral functions. Methods: A literature search was conducted in PubMed and the Cochrane Library. Studies relating to preterm infants (gestational age <37 weeks) were included in the analysis. Retrieved papers were categorized according to their main topic: CF timing and quality; clinical outcome; recommendations; strategies in infants with oral dysfunction. Results: The literature search in PubMed retrieved 6295 papers. Forty met inclusion criteria. The Cochrane search identified four additional study protocols, two related to studies included among PubMed search results, and two ongoing trials. Moreover, among 112 papers dealing with oral feeding, four aiming at managing CF in preterm infants with oral dysfunctions were identified. Conclusions: The available literature does not provide specific guidelines on the management of CF in preterm infants, who are generally weaned earlier than term infants. There is a paucity of data regarding the relationship between CF and growth/quality of growth and health outcomes in preterm infants. It could be suggested to start CF between five and eight months of chronological age if infants have reached three months corrected age and if they have acquired the necessary developmental skills. An individualized multidisciplinary intervention is advisable for preterm infants with oral dysfunctions.

## 1. Introduction

Optimal nutrition in the first 1000 days, from conception to the second year of life, has the potential to shape individual health status during both childhood and adult life [1]. The relationship between nutrition in early life and long-term outcome is particularly relevant for preterm infants, whose intrinsic immaturity makes nutritional management a daily challenge for the neonatologist.

Despite the fact that scientific interest in long-term effects of preterm infants’ nutrition is constantly growing, to date very little attention has been paid to complementary feeding (CF), also known as weaning, which is defined by the World Health Organization as “the process starting when breast milk alone is no longer sufficient to meet the nutritional requirements of infants’’ so that ‘‘other foods and liquids are needed, along with breast milk’’ [2]. No guidelines about CF in preterm infants exist, and both the recent recommendations issued by the European Society of Pediatric Gastroenterology, Hepatology and Nutrition (ESPGHAN) [3] and the Italian Society of Pediatrics [4] are specifically intended to guide the introduction of solid foods to “healthy term infants”. Therefore, clinicians’ attitude towards CF in preterm infants is extremely variable both within and among countries [5]; actually, a recent survey, promoted by the Italian Society of Pediatrics and conducted among primary care pediatricians, has documented a wide variability in timing of CF introduction and type of foods proposed to start CF [6]. 

Thus, the aim of the present paper is to analyze systematically the available literature about current practices regarding the timing and characteristics of the introduction of CF among preterm infants (defined as infants born before 37 weeks gestational age [GA]). Furthermore, we aimed to investigate the available literature regarding the management of CF in preterm infants who have developed comorbidities that may interfere with oral functions.

## 2. Methods

### Literature Search

The study protocol was designed jointly with the members of the Study Group on Nutrition of the Italian Society of Neonatology.

A systematic review of published studies dealing with CF in preterm infants was performed, following PRISMA guidelines [7]. Studies providing information about all preterm infants, defined as those born before 37 weeks GA, were included. No limitation regarding study design was applied, in order to include all the available literature. A search was conducted in PubMed (http://www.ncbi.nlm.nih.gov/pubmed/) for studies published before 10th April 2020, using a search string developed ad hoc for this purpose. This string was built up by combining all the terms related to CF in preterm infants, using PubMed MeSH terms, free-text words, and their combinations, through the most proper Boolean operators, in order to be as comprehensive as possible (Figure 1). A similar search strategy was used to search the Cochrane Library (https://www.cochranelibrary.com/advanced-search).

The search was conducted by AA and NL: relevant studies were identified from the abstract, and reference lists of papers retrieved were searched for additional studies. “Snowballing” technique was also used [8].

## 3. Results

### Literature Search

The literature search in PubMed retrieved 6295 papers, for which we examined the title and/or the abstract. Among these papers, and according to the title and/or abstract, 40 apparently dealt with or included information about CF practices in preterm infants; for these papers, the full text was evaluated. As shown in Figure 1, the remaining 6255 were excluded for the following reasons: 6129 did not actually deal with CF, 14 were focused on nutrient supplementation, and 112 dealt with promotion of oral feeding. 

After examining the full texts of the 40 retrieved papers, 14 were excluded, as 9 did not actually deal with CF and 5 reported data only about term infants, leading to 26 suitable studies [5,6,9,10,11,12,13,14,15,16,17,18,19,20,21,22,23,24,25,26,27,28,29,30,31,32]. Five additional papers were identified from the reference lists of included studies [33,34,35,36,37]; 31 studies were then included in the systematic review [5,6,9,10,11,12,13,14,15,16,17,18,19,20,21,22,23,24,25,26,27,28,29,30,31,32,33,34,35,36,37].

Among these, there were six narrative reviews [5,14,15,22,27,32], one systematic review [13], two meta-analyses [11,33], one commentary [17], two recommendations [36,37] and one study protocol [31]. Among the remaining 18 trials, there were 14 observational studies [6,9,10,12,18,19,20,21,24,28,29,30,34,35], two randomized controlled trials (RCTs) [16,25] and two papers pooling data from different RCTs [23,26].

The search performed in the Cochrane Library retrieved 757 papers (47 Cochrane reviews, 2 Cochrane protocols, and 708 trials). Among papers suitable for inclusion in the systematic review, there were four papers duplicating the PubMed results [16,23,25,26]. The Cochrane search identified four study protocols: two of them were related to studies already included among PubMed search results [13,16]; the remaining two trials are ongoing (study IDs: ClinicalTrials.gov Identifier: NCT01809548 and CTRI/2012/04/002572).

Papers excluded from the primary search as dealing primarily with oral feeding were examined in detail, to evaluate whether they reported any specific advice for CF in infants with feeding dysfunction. Among these 112 papers, only four actually aimed at managing CF in preterm infants with oral dysfunctions [38,39,40,41]. Of the remaining 108 papers, three aimed at investigating breastfeeding rates in preterm infants, one dealt with weaning from parenteral to enteral nutrition, 82 concerned preterm infants’ feeding issues in terms of achievement of autonomous sucking, whereas 21 papers aimed at evaluating the scales for measuring oral skills in preterm infants.

Figure 1 shows the search string and the flow chart depicting the search strategy with relevant included and excluded studies. Reasons for study exclusion are also reported. Relevant characteristics of included studies are reported in Table 1. 

## 4. Discussion

### 4.1. Timing and Quality of Complementary Feeding

Most data regarding CF in preterm infants are reported in observational trials: despite wide variability in timing and quality of CF, all the available studies document that preterm infants are introduced early to solid foods. Data from a recent cross-sectional study performed in Poland and Austria show that preterm birth is associated with a four-to-tenfold increase in the risk of being introduced early to solid foods compared to term birth, with almost 60% of preterm infants in Austria and 80% in Poland receiving CF before four months of age [9].

Even when taking account of their corrected age (CA), preterm infants are generally introduced to solid foods earlier than term infants. In the study by Norris et al., performed in the UK in 2003, 95% of the included preterm infants were weaned very early (mean age 11.5 weeks CA) [28]. Similarly, a recent study performed in Australia documented that preterm infants started CF at a median age of 14 weeks CA, compared to a median age of 19 weeks in term controls [10]. 

Data from a survey conducted in Italy in the early 2000s show that more than 60% of preterm infants were weaned before four months CA; in most cases, the first solid food was nutritionally inadequate, with a low energy and protein content [21]. Similarly, data collected in the US document that almost half of the studied preterm infants received some solid food before four months CA, and this probably contributed, in that population, to a rapid increase in weight gain, without a similar increase in infants’ length [34]. 

The degree of prematurity constitutes a major determinant in the timing of CF introduction: in the study by Braid et al., compared to term infants, preterm infants born between 22 and 32 weeks GA had a 9.90 odds of receiving solid food before four months of age; the odds in moderately and late preterm infants was 6.19 [20].

Very recent data from a survey conducted in Italy among primary care pediatricians confirm all the findings above, and document a wide variability in the timing of introduction of CF, in the type of offered solid foods and in vitamin and iron supplements prescribed during the CF period to Italian preterm infants. The authors found that the type of complementary foods did not comply with an evidence-based sequence; 98% of the pediatricians promoted vitamin D supplementation and 89% promoted iron supplementation, with great diversity in timing and doses. [6]. Data from an observational study performed in Italy suggest that late preterm (GA 34–36 weeks) infants are also introduced to solid foods earlier than recommended, at a mean postnatal age of 5.7 months and a mean CA of 4.6 months [29]. In this cohort, specific feeding practices were associated with huge variation in the timing of CF: specifically, the introduction of fruit as the first type of solid food was associated to a 1.4 months earlier CF initiation. 

The tendency towards an early introduction of CF in preterm infants might be attributable at least partially to the lack of appropriate nutritional education given to family members of preterm infants regarding CF and its potential consequences on short- and long-term health. A recent Cochrane review was aimed at investigating whether providing education to family members regarding CF would improve growth and development of preterm infants. However, no eligible trials were found, and this is probably related to the scarce evidence in identifying the ideal weaning strategy for premature infants [11]. In this respect, a survey conducted in the UK pointed out that many mothers of low birthweight infants misjudged a high-fiber, low-fat diet as nutritionally appropriate for their infants, and did not perceive an adequate amount of calories as nutritionally important [35].

Only few trials have investigated in a randomized setting differences between specific timings of CF introduction: in the paper by Marriott et al., published in 2003, sixty-eight preterm infants were randomly assigned to a “preterm weaning strategy (PWS)” group or to a control group. Infants in the PWS group received semisolid food as soon as 13 weeks of age, provided they had reached at least 3.5 kg body weight. Parents of infants in the PWS group were advised to use high-energy and high-protein foods, and to mix dried cereals and home-prepared foods with a preterm infant formula. Infants in the control group were started on CF at 17 weeks of age, provided they weighted at least 5 kg, and no specific advice for food quality was given. At 12 months of age, infants in the PWS group had greater length compared to those in the control group, with no differences in weight or head circumference. However, it is unclear whether improved length could be attributed exclusively to an early CF or to a combination of early CF and recommendations on food energy and protein content [25]. A recent open-label, randomized trial conducted in three public health facilities in India compared two timings of CF in infants with GA <34 weeks: while there was no difference in the primary outcome (weight-for-age Z score at 12 months CA) between infants receiving CF at 4 vs. 6 months CA, preterm infants weaned at four months CA needed hospital admission more frequently compared to those weaned at six months CA [16]. Due to the specific setting where the study was conducted, the generalizability of these data is uncertain, but the results of the study point out the need for a more focused attention not only on the timing, but also on the quality of solid foods offered to preterm infants [17].

### 4.2. Complementary Feeding and Clinical Outcome

Few studies investigated the potential relationship between the timing for CF introduction and long-term consequences on preterm infants’ health. 

As for growth, data from five trials conducted in the UK in the mid-90s and including over 1600 term and preterm infants showed that weaning as early as 12 weeks of chronological age could increase early weight gain, but that long term growth does not appear to be significantly different between infants receiving CF before or after this age [23]. Similarly, Spiegler et al. did not document any negative effect of early introduction of solid food on height and weight at two years of age in almost 1000 former preterm infants in Germany [19].

At present, there are insufficient data to clarify whether variations in the timing of CF introduction could also relate to an increase in obesity and overweight in preterm infants [13] as reported for full term infants. Indeed, in full term infants, untimely introduction of solid foods, either before four months of age [42] or later than six–seven months of age [43], might be associated with higher adiposity and increased risk of overweight and childhood obesity. 

As for allergy and atopic manifestations, a retrospective observational study performed in Finland compared timing of CF introduction in term and preterm infants, showing that the highest degrees of prematurity were associated with the earliest introduction of CF (less than 1 month CA in extremely preterm infants); however, the incidence of food allergies and atopic dermatitis did not differ between preterm and term infants by the ages of one and two years, suggesting that early exposure to solid foods in preterm infants might have no detrimental consequences in terms of allergic disease [12]. These data are in contrast with the results from the study by Morgan et al., who showed that preterm infants introduced to at least four solid foods by 17 weeks CA had a high risk of developing eczema by 12 months of age [24]. 

Given the paucity of available data, at present no specific recommendation can be made on the optimal timing of CF in preterm infants in relation to growth and health issues later in life.

### 4.3. Currently Available Recommendations on CF

While no guidelines on CF introduction in preterm infants are available at present, a few weak recommendations exist, most of which have been issued in the UK by dieticians’ working groups and professional associations. Some brief advice about preterm infants was first included in a Report published in 1994 by the “Committee on Medical Aspects of Food and Nutrition” (COMA report [37]): the authors suggested that preterm infants should be weaned to solid foods once they had reached a minimum body weight of 5 kg, and that CF should be started once the infant had acquired a few specific developmental milestones. 

The suggestions made in the COMA report, however, apply poorly to “modern” populations of preterm infants: actually, the choice of a weight-limit for starting CF in preterm infants is probably misleading, as infants born earliest and growing slowly would reach 5 kg well beyond the optimal temporal window (four to six months of age) recommended by the ESPGHAN to start CF in term infants. In addition, the choice of selected developmental milestones to be reached before beginning CF would not consider the overall infant development, thus potentially leading to further delay.

To overcome the limitations of the criteria proposed in the COMA report, it has been suggested that CF should be started in preterm infants within a specific temporal window, between five and eight months of uncorrected age [36]. The choice of such a window strings together behavioral and developmental considerations, including the overall infant’s oro-motor development, the acquisition of taste, and the readiness to explore new textures. Actually, between five and eight months of age, virtually all preterm infants should have acquired the developmental skills which allow the consumption of foods other than milk, such as the progressive disappearance of the protrusion reflex of the tongue, the reduction of reflexive suck in favor of lateral tongue movements, and the gradual appearance of lip seal. In addition, this time window is the optimal one in term infants for introducing new flavors and textures: even if it is not known how this sensitive period is affected by preterm birth, it is highly likely that the later preterm infants are introduced to new flavors and textures, the less likely they are to accept a wide variety of foods [5,36]. A single study compared emotional responses of preterm and term infants at their first contact with solid foods: contrary to the authors’ expectations, preterm infants showed fewer negative emotions in response to novel food and seemed to adapt rapidly to solid food, with a reduced frequency of negative facial expressions after repeated exposure to novel foods [18].

Safe and successful transition to solid food must rely on an adequate motor development, which is hardly achieved by preterm infants before three months CA. For this reason, it seems reasonable to also include CA in the evaluation of the optimal timing for starting CF in preterm infants, as this would also constitute a unifying criterion for the widely heterogeneous population of preterm infants, being applicable to both extremely and late preterm infants. 

### 4.4. Complementary Feeding Strategies for Infants with Oral Dysfunctions

Infants born preterm are at high risk to develop oral dysfunction [44,45]. The occurrence of comorbidities, such as bronchopulmonary dysplasia (BPD) or neurodevelopmental impairment can hinder the physiological development of oral skills, and some authors reported that more than 15% of preterm infants were discharged on enteral feeding (tube feeding) [46]. It was reported that infants born before 30 weeks showed a higher risk of developing oro-motor feeding problems at 12 months’ CA than their term-born peers, especially if they had undergone neonatal surgery [40].

The need for tailored recommendations regarding CF is suggested by a recent trial by Menezes et al. showing that defensive behaviors at mealtime are common in preterm infants starting CF. among the investigated 38 infants, 42% refused to open their mouth, 29% showed food selectivity and 26% experienced feeding refusal. A potential association between formula feeding and risk of feeding refusal was also suggested [30].

A recent editorial comment proposed the use of paladai for preterm infants who have developed oral dysfunctions [38]. The paladai is a traditional feeding device used in some Indian communities that consists in feeding infants with soft consistency food through a special spout or with a spoon. Indeed it has been reported that preterm infants are able to ingest a greater amount of food by using a spoon-assisted mode of feeding [47], probably due to the decreased gag reflex elicited by the introduction of food with higher texture [41]. In addition, in a prospective study conducted on 40 preterm infants, evaluated during the first post menstrual month, and then at 6, 9, 12 and 24 months, the authors showed that the quality of sucking patterns was not associated with skill level achievement of assisted spoon feeding and that the food consistency offered and the length of feeding experience influenced the acquisition and quality of oral motor skills [39].

Preterm infants who developed oral dysfunctions need a multidisciplinary follow up that must include a nutritionist and a speech therapist specializing in oral function, in order to individualize the intervention and achieve, partially or totally, oral feeding [41,48].

Nowadays, there are no guidelines regarding CF strategies for preterm infants that developed major comorbidities, therefore the nutritional management for these infants should be individualized before discharge and should be revised regularly. Infants with associated gastrointestinal problems need a follow up by a pediatric gastroenterologist. Low salt, and limited volume, high calorie nutrition may be required for infants with BPD; these infants also can become slightly hypoxic on feeding and often tolerate foods given by spoon better than liquids given by nipple. Spoon feeding may begin at three to four months of corrected GA. Foods with thicker consistency may be swallowed more easily. With volume restriction, high-calorie solid foods could be offered. Oro-motor stimulation should be started for infants needing prolonged tube feeds as soon as possible. Many feeding problems may be related to psychosocial issues and referral to a behavioral psychologist may be required [49].

Clinical experience suggests that using complete foods, based on amino-acid mixtures in non-allergic infants can also ensure the fulfilment of the high nutritional needs of infants that developed comorbidities. These products have the characteristic of concentrating in small volumes a high macronutrients content. Their neutral taste, possibly flavored with milk or fruit, makes them also well tolerated in the youngest infants that are not able to consume, even with a spoon, large quantities of weaning food. Therefore, it is possible to create a “modern paladai” to guarantee high-calorie/protein intakes in those infants with oral feeding issues before 3-4 months CA for weaning these infants under the strict monitoring of nutritionists and speech therapists specialized in oral function.

## 5. Conclusions

Nutrition in early life has the potential to influence preterm infants’ clinical outcomes both in the short and in the long term. At present, it is unclear which is the specific contribution of CF in shaping preterm infants’ health. The results of this systematic review highlight a paucity of data on the optimal characteristics of solid food introduction in infants born preterm. 

The available literature does not provide strong recommendations on the management of CF in preterm infants. It has been suggested that preterm infants should introduce solid foods between five and eight months of chronological age, provided they have reached at least three months CA. However, available data suggest that there is no specific timing which applies safely to all preterm infants, who constitute an extremely variable population in terms of achievement of developmental milestones and oral skills. For this reason, we propose that CF in preterm infants should be introduced in preterm infants following an individualized evaluation mainly based on infants’ development rather than corrected or postnatal age. Furthermore, preterm infants who have developed oral dysfunction would need an individualized intervention conducted by a multidisciplinary team that must include clinicians, nutritionists and speech therapists specialized in oral function. Further studies, preferentially intervention studies or randomized controlled trials, including preterm infants categorized by gestational age at birth, and aiming at evaluating growth, quality of growth and health in the short and long-term in relation to CF practices, are advisable. 

## Figures and Tables

**Figure 1 nutrients-12-01843-f001:**
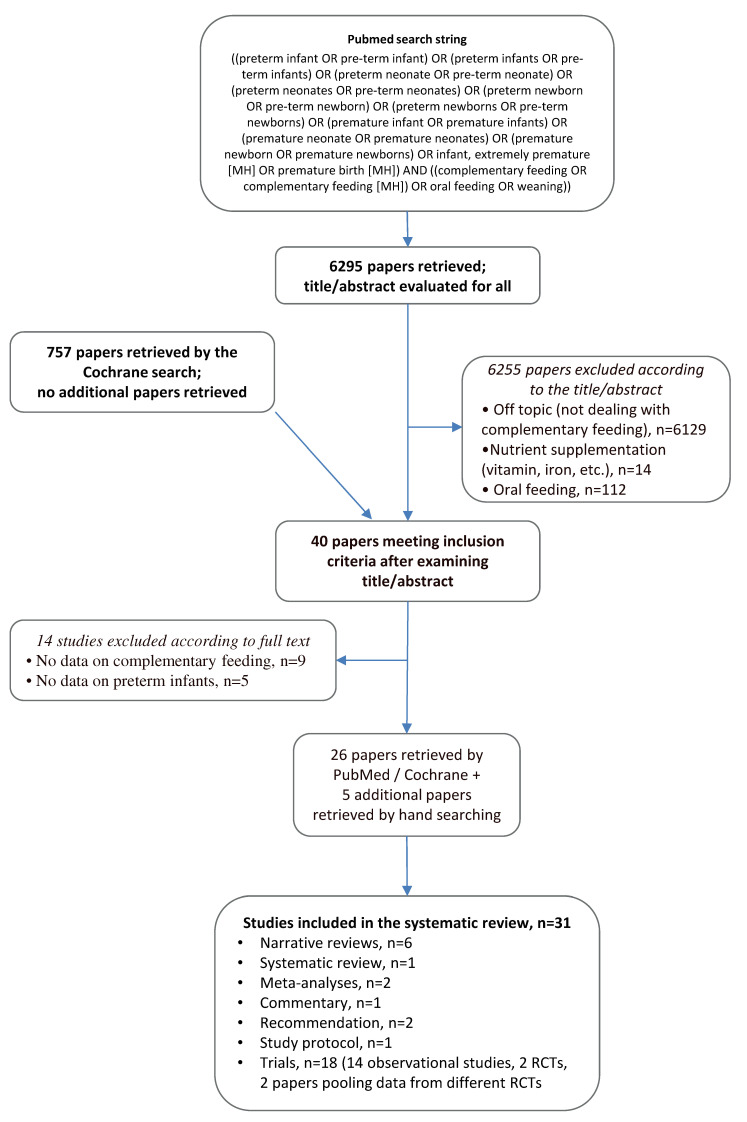
Search string and flow chart depicting the search strategy with relevant included and excluded studies. Reasons for study exclusion are also reported.

**Table 1 nutrients-12-01843-t001:** Main characteristics of the studies included in the systematic review are described. Studies are classified according to study design: recommendations, reviews, and trials. CA: corrected age; CF: complementary feeding; PCF: preterm complementary feeding; RCT: randomized controlled trial.

Author, Year	Study Design	Main Findings
**Recommendations**		
COMA report, 1994 [37]	Recommendation	Report of the Working Group on the Weaning Diet of the Committee on Medical Aspects of Food Policy. Recommendations for CF in term infants. Brief specific advice for preterm infants.
King, 2009 [36]	Recommendations	Proposed evidence-based guide for CF tailored to preterm infants.
**Reviews**		
Barachetti, 2018 [15]	Narrative review	Review of management, timing and health outcomes of CF in preterm infants.
Embleton, 2017 [17]	Commentary on Gupta, 2017	Commentary on the trial by Gupta et al. examining the effect of two different timings of PCF on preterm infants’ growth and clinical outcome
Fanaro, 2007 [22]	Narrative review	Review of growth and feeding issues in preterm infants after hospital discharge, with specific focus on age, type and frequency of complementary foods
Fanaro, 2009 [32]	Narrative review	Review of CF introduction in preterm infants
Foote, 2003 [27]	Narrative review	Overview of CF with a focus on low birth weight and preterm infants
Palmer, 2012 [5]	Narrative review	Available guidelines and current practices regarding PCF; evaluation of possible harms of early introduction of solid foods
Peters, 2018 [14]	Narrative review	Review on nutrition after discharge of preterm infants from the Neonatal Intensive Care Unit, with a focus on CF and allergenic foods
Elfzzani, 2019 [11]	Meta-analysis	Role of nutritional education of family members in supporting CF practices in preterm infants. No eligible trials looking at the impact of nutrition education of family members on PCF fulfilled the inclusion criteria of this systematic review.
Gupta, 2016 [33]	Study protocol for a meta-analysis	To evaluate the effect and safety of early (at or before four months) vs. late (after four months) initiation of PCF. Both corrected and postnatal age will be examined.
Vissers, 2016 [31]	Study protocol for Vissers, 2018 [13]	Protocol for a systematic review on the effect of PCF timing on overweight.
Vissers, 2018 [13]	Systematic review	Effect of the timing of CF introduction (early vs. late) on the risk of overweight in preterm infants The five included papers (thee RCTs, two cohort studies) showed conflicting results: two RCTs → no significant difference in BMI Z-score between the intervention groups at 12 months of age; one RCT → higher rate of length growth until 12 months in the preterm weaning strategy-group compared with the current best practices; one observational study → inverse relationship between timing of CF and length and weight Z-scores.
Chawla, 2019 [38]	Editorial commentary	An overview regarding the CF strategies for preterm infants with bronchopulmonary dysplasia
Dusick, 2003 [41]	Narrative review	A review regarding the nutritional management of infants with dysphagia.
**Trials**		
Baldassarre, 2018 [6]	Observational trial	Survey of PCF among Italian primary care pediatricians. Heterogeneity in PCF timing (based on infants age, and/or neurodevelopment and/or body weight), quality, and prescription of vitamin D and iron supplements.
Braid, 2015 [20]	Observational trial	Analysis of factors associated with early CF in preterm infants from the Early Childhood Longitudinal Study, Birth Cohort (2001–2002). Higher odds of early CF in preterm vs. term infants. The lower the GA, the higher the odds. Predictors of early CF different in preterm compared to term infants.
Cleary, 2020 [10]	Observational trial	Structured interviews on infant feeding practices, growth and medical status in term and preterm infants. Preterm infants received CF earlier than term infants; lower maternal education and male gender were associated with early CF among preterm infants.
Fanaro, 2007 [21]	Observational trial	Survey of CF practices in an Italian region. Wide variation in timing (corrected vs. chronological age) and quality of CF (low energy and low protein often offered as first solid food, with negligible iron and zinc content).
Fewtrell, 2003 [26]	Pooled RCTs results	Data from >2000 infants from seven prospective UK RCTs, comparing the age at CF in term appropriate size for gestational age (AGA), small for gestational age, and preterm infants. Preterm infants were significantly more likely to receive solids at both six and twelve weeks after term than term AGA infants. Factors associated with earlier CF were formula feeding and maternal smoking.
Giannì, 2018 [29]	Observational trial	Evaluation of practices related to CF in a cohort of Italian late preterm infants. Late preterm infants were weaned at almost six months of age and received low energy and/or low protein-dense foods as first solid foods.
Gupta, 2017 [16]	RCT	RCT comparing the initiation of CF at four vs. six months CA in preterm infants in India. No difference was documented in the primary aim (weight-for-age z score at 12 months CA) between groups, but a higher rate of hospital admission was recorded in the four-month group.
Hüb, 2020 [39]	Observational trial	Evaluation of the association between sucking patterns, assisted spoon feeding, and chewing skills in preterm infants. Sucking patterns were evaluated at 34, 37, and 44 weeks CA, assisted spoon feeding was evaluated at six, nine, and 12 months PMA, and chewing was evaluated at 9, 12, and 24 months PMA.
Longfier, 2016 [18]	Observational trial	Analysis of facial expression and infant’s temperament in response to the introduction of CF in preterm vs. term infants. Infants born preterm expressed fewer negative emotions in response to first CF than infants born full-term and showed a familiarization effect with the frequency of negative expressions decreasing after tasting the second spoon, regardless of infant age, type of food and order of presentation.
Marriott, 2004 [25]	RCT	RCT aimed at comparing a “preterm weaning strategy (PWS)” vs. conventional CF management in preterm infants. Infants receiving CF according to PWS showed higher standard deviation length scores and length growth velocity, and higher intake of energy, protein, and carbohydrate, and iron during follow up.
Menezes, 2018 [30]	Observational trial	Structured interviews administered to parents of preterm infants to highlight feeding difficulties during CF. Most infants had at least one defensive behavior at mealtime, including refusal to open their mouth, food selectivity, and feeding refusal.
Morgan, 2004 [23]	Pooled RCTs results	Data from >1600 term and preterm infants from five prospective UK RCTs, comparing early (<12 weeks) vs. late (>12 weeks) introduction of CF. As for preterm infants, those weaned before 12 weeks showed slower gain in weight, length, and head circumference between 12 weeks and 18 months than those weaned after 12 weeks; by 18 months, there were no significant differences in size between the two groups. No effect of CF on other clinical outcomes was observed
Morgan, 2004 [24]	Observational trial	Evaluation of CF-related risk factors for eczema at 12 months post-term in preterm infants. Identified risk factors were the introduction of ≥4 solid foods by or before 17 weeks post-term, male gender, having atopic parents who introduced solid foods before 10 weeks post-term or having at least one atopic parent.
Morgan, 2016 [35]	Observational trial	Cohort study performed by means of postal questionnaires aimed at describing feeding patterns and mothers’ perceptions of desirable feeding practices in preterm infants in England. CF was introduced at a median age of 17 postnatal weeks. Mothers perceived a high-fiber, low-fat diet as important for their infants. A high calorie intake was not given the correct importance by 25% of mothers.
Norris, 2002 [28]	Observational trial	Structured interviews conducted in the UK to evaluate factors associated with PCF. Almost half of the infants received early CF, both considering corrected and chronological age. Differences between human milk- and formula-fed infants in the timing of CF were documented.
Rodriguez, 2018 [34]	Observational trial	Cross-sectional study aimed at examining the relationship between feeding practices and weight gain. Almost half infants received CF before four months CA. A greater weight gain was documented in infants receiving early CF, but the results were considered of little clinical relevance.
Sanchez, 2016 [40]	Observational trial	Evaluation of oro-motor feeding at 12 months’ CA in children born before 30 weeks’ GA compared with term-born peers by observational assessment. Infants born before 30 weeks presented with higher odds of oro-motor feeding problems at 12 months’ CA than their term-born peers (OR 2.21; 95% CI 1.55–3.16). Neonatal surgery was associated with increased odds of feeding difficulties in children born before 30 weeks (OR 11.66; 95% CI 1.56–87.23).
Spiegler, 2015 [19]	Observational trial	Longitudinal analysis of timing of CF introduction in German VLBW infants, risk factors for early introduction of CF, and relationship between PCF timing and growth at 2 years of age. Average age at introduction of CF: 3.5 months post-term. Low GA at birth = early PCF introduction. Age at introduction of CF influenced by intrauterine growth restriction, GA at birth, maternal education and a developmental delay perceived by the parents. No negative effect of early introduction of CF on length and weight at two years of age.
Yrjänä, 2018 [12]	Observational trial	Evaluation of the association between very early introduction of semi-solid foods on food allergies or atopic dermatitis. Preterm infants were introduced safely to semi-solid foods earlier than term infants but did not show an increased risk for food allergies or atopic dermatitis.
Zielinska, 2019 [9]	Observational trial	Cross-sectional study investigating factors for early CF in Poland and Austria. Preterm birth was identified among significant risk factors for early CF, together with lower maternal age and educational level, absence of breastfeeding and formula feeding after hospital discharge.

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
