# Peer review of "Complementary Feeding in Preterm Infants: A Systematic Review"

_nutrients, 2020, doi:10.3390/nu12061843_

Round 1

Reviewer 1 Report

Study: Complementary feeding in preterm infants: a systematic review

Design: Systemic review

Abstract: The abstract defines the background and methods with a snapshot of conclusion- it may be easier to read if bulleted the information:

Background:

Methods:
Results:

Conclusion

In an easier to read format.

Also, please identify what was exclusion and clearer inclusion (missing data, age categories?, definition of preterm)

Introduction: please identify preterm infant (<37 weeks?) Late preterm? Early preterm, all?

Methods: While the robust search is appreciated on preterm – please define -i.e., infants<37 weeks; also it would be very interesting to sub categorize by birth GA if there was correlation to when CF started

Results: Please identify your definition of preterm; also would be extremely interesting to clinicians if you would develop a table of GA; i.e. GA 23-26 weeks 26-28, 29-30, 31-33, etc to see if any differences or trends that would be correlated to timing and success of CF.

Also any differences in type of CF; i.e iron strained meats> cereals?

Discussion:

First line of page 8 appears to be missing number 00s?

Any discussion on type of food would be valuable and impact on Vit D status, iron deficiency anemia outcomes?

Author Response

Reviewer 1

Study: Complementary feeding in preterm infants: a systematic review

Design: Systemic review

Abstract: The abstract defines the background and methods with a snapshot of conclusion- it may be easier to read if bulleted the information:

Background:

Methods:
Results:

Conclusion

In an easier to read format.

We thank the Reviewer for this comment. According to his/her suggestion, the abstract has been modified by structuring the different sections as follows:

“Background: This systematic review summarizes available literature regarding complementary feeding (CF) in preterm infants, with or without comorbidities that may interfere with oral functions.

Methods: A literature search was conducted in PubMed and the Cochrane Library. Studies relating to preterm infants (gestational age <37 weeks) were included in the analysis. Retrieved papers were categorized according to their main topic: CF timing and quality; clinical outcome; recommendations; strategies in infants with oral dysfunction.

Results: The literature search in PubMed retrieved 6295 papers. Forty met inclusion criteria. The Cochrane search identified 4 additional study protocols, 2 related to studies included among PubMed search results, and 2 ongoing trials. Moreover, among 112 papers dealing with oral feeding, 4 aiming at managing CF in preterm infants with oral dysfunctions were identified.

Conclusions: The available literature does not provide specific guidelines on the management of CF in preterm infants, who are generally weaned earlier than term infants.

There is paucity of data regarding the relationship between CF and growth/quality of growth and health outcomes in preterm infants. It could be suggested to start CF between 5 and 8 months of chronological age, if infants have reached 3 months corrected age and if they have acquired the necessary developmental skills. An individualized multidisciplinary intervention is advisable for preterm infants with oral dysfunctions.”

Also, please identify what was exclusion and clearer inclusion (missing data, age categories?, definition of preterm)

With regards to inclusion and exclusion criteria, we aimed to be as comprehensive as possible, including in our systematic review papers with different study designs to analyze all the available information on this topic.

The literature search using the ad hoc search string actually retrieved a huge number of papers which were classified as “off-topic”, as dealing with various aspects of infants’ nutrition different from complementary feeding (reasons for exclusion are detailed in Figure 1).

Definition of preterm birth (GA <37 weeks) has been specified as requested throughout the manuscript.

Introduction: please identify preterm infant (<37 weeks?) Late preterm? Early preterm, all?

We agree with the Reviewer’s comment, the definition of preterm birth has been added also into this section.

Methods: While the robust search is appreciated on preterm – please define -i.e., infants<37 weeks; also it would be very interesting to sub categorize by birth GA if there was correlation to when CF started

Information about GA inclusion criteria was added in the methods section as following:

“Studies providing information about all preterm infants, defined as those born before 37 weeks gestational age, were included”

We agree with the Reviewer that it would be extremely interesting to sub categorize infants by GA, in order to provide more specific information on CF for each group of preterm infants: unfortunately, however, the available literature on CF in preterm infants did not allow us to give recommendations categorized for GA. We tried to review available studies in this perspective, but information was limited and conflicting. We detailed the available information in table 1 and in the discussion but were unable to draw specific recommendations according to age groups.

Results: Please identify your definition of preterm; also would be extremely interesting to

clinicians if you would develop a table of GA; i.e. GA 23-26 weeks 26-28, 29-30, 31-33, etc to see if any differences or trends that would be correlated to timing and success of CF.

We thank the Reviewer for this suggestion. Unfortunately, as stated before, we did not perform an analysis categorized for GA due to the paucity of data classified according to GA. Actually, in some studies the authors did not specify the definition for prematurity and in other studies infants were included according to birth weight rather than GA.

 Also any differences in type of CF; i.e iron strained meats> cereals?

Among studies included in our systematic review, we found a recent survey conducted in Italy among primary care paediatricians regarding the time of introduction of CF, the type of offered solid foods and the vitamin and iron supplements prescribed during the CF period to Italian preterm infants. The authors found that the type of complementary foods did not comply with an evidence-based sequence; 98% of participants promoted vitamin D supplementation and 89% promoted iron supplementation with great diversity in timing and doses. More detailed information about this survey is now provided in the Discussion section.

Unfortunately, no other specific information about the quality of foods offered during CF is available.

Discussion:

First line of page 8 appears to be missing number 00s?

We modified the number into 2000s.

Any discussion on type of food would be valuable and impact on Vit D status, iron deficiency anemia outcomes?

We thank the Reviewer for this suggestion. In the recent Italian survey about complementary feeding, the authors investigated the time of introduction of CF, the type of offered solid foods and the vitamin and iron supplements prescribed during the CF period to Italian preterm infants. The authors found that the type of complementary foods did not comply with an evidence-based sequence; 98% of participants promoted vitamin D supplementation and 89% promoted iron supplementation with great diversity in timing and doses. This information is now provided in the discussion section.

Reviewer 2 Report

Well written paper. Interesting. 

One comment: the authors make quite affirmative recommendations in the abstract regarding the start of complementary feeding while they stat that there is paucity of data. 

As as consequence, i do think that the recommendation so be formulated more carefully. They are more careful in the conclusion of the paper than in the abstract. But many people read only the abstract. 

It would be nice if the authors could add recommendations regarding what design would give the best possible answer to the relevant questions. 

Author Response

Reviewer 2

Well written paper. Interesting. 

One comment: the authors make quite affirmative recommendations in the abstract regarding the start of complementary feeding while they stat that there is paucity of data. 

As consequence, i do think that the recommendation so be formulated more carefully. They are more careful in the conclusion of the paper than in the abstract. But many people read only the abstract. 

We thank the Reviewer for this suggestion. The abstract conclusions have been modified accordingly.

It would be nice if the authors could add recommendations regarding what design would give the best possible answer to the relevant questions. 

We thank the Reviewer for this precious comment. The study conclusions have been modified following this comment.